# Novel Modelling Approach for the Calculation of the Loading Performance of Charging Stations for E-Trucks to Represent Fleet Consumption

**Thomas Märzinger** [1] , **David Wöss** [1,*] , **Petra Steinmetz** [2] , **Werner Müller** [1] **and Tobias Pröll** [1]

1   Institute of Chemical and Energy Engineering, University of Naturel Resources and Life Sciences, 1190 Vienna, Austria; thomas.maerzinger@boku.ac.at (T.M.); werner.mueller@boku.ac.at (W.M.); tobias.proell@boku.ac.at (T.P.)
2   VOIGT+WIPP Industrial Research GmbH, 1150 Vienna, Austria; steinmetz@voigt-wipp.com
*   Correspondence: david.woess@boku.ac.at; Tel.: +43-1-47654-89316

**Abstract:** In its "Sustainable and Smart Mobility Strategy", the European Commission assumes a 90% reduction in traffic emissions by 2050. The decarbonisation of transport logistics as a major contributor to climate change is, therefore, indicated. There are major challenges in converting logistic transport processes to electric mobility. Currently, there is little available information for the conversion of entire fleets from fossil to electric fuel. One of the biggest challenges is the additional time needed for recharging. For the scheduling of entire logistics fleets, exact knowledge of the required loading times and loading quantities is essential. In this work, a parametrized continuous function is, therefore, defined to determine the essential parameters (recharging time, retrieved power, energy amounts) in HPC (high-power charging). These findings are particularly important for the deployment of multiple e-trucks in fleets, as logistics management depends on them. A simple function was constructed that can describe all phases of the charging process in a continuous function. Only the maximum power of the charging station, the size of the battery in the truck and the start SOC (state of charge) are required as parameters while using the function. The method described in this paper can make a significant contribution to the transformation towards electro-mobile urban logistics fleets. The required charging time, for example, is crucial for the planning and scheduling of e-logistics fleets and can be determined using the function described in this paper.

**Keywords:** logistics; e-mobility; mathematical function; high-power charging; fleet conversion; e-truck; charging function

## 1. Introduction

The European Green Deal assumes a massive reduction in greenhouse gas emissions in the transport sector. A reduction in emissions of 55% by the year 2030 is announced [1]. The way to achieve this is by increasing sustainability in the transport sector. Various LCA (life cycle assessment) analyses for the field of urban logistics show that the use of e-trucks can contribute significantly to the reduction in greenhouse gas emissions [2–4]. Transport logistics, in particular, will face special challenges here [5].

The time needed for charging is seen as especially problematic for planning in logistical processes [6,7]. The accurate simulation of charging processes in electro mobility is becoming increasingly important as the number of vehicles increases, particularly in the field of urban logistics, where numerous vehicles must be recharged simultaneously at a hub [8].

This paper was developed in the course of the "MegaWatt Logistics" research project [9]. The aim of the project is to develop strategies based on a field test with 8 e-trucks (26t, MAN, Steyr, Austria) in order to force a fleet to convert to electric mobility. The project showed that the exact knowledge of the charging times, as well as the energy quantities, are crucial for the management of e-trucks.

If the charging processes are simultaneous, an imprecise representation of the recharging performance leads to high errors due to the superimposition of the individual processes. Therefore, the total power consumption and the implication on the power supply system can only be predicted insufficiently.

In addition to the instantaneous power required on site, knowledge of the charging quantity and time required for the tours is decisive for the dispatching of e-fleets. For practical use, therefore, a function is needed that maps the charging process as simply and as accurately as possible. These findings form the basis for an efficient use of e-vehicles in current logistics management tools.

### 1.1. Literature Review

A wide range of charging pole models are currently cited in the literature. On the one hand, complex prediction models for the exact determination of the superposition of individual charging processes are discussed [10,11].

The simpler models usually use a constant power supply [12,13]. This is a simple approach that gives a good match with reality when charging at low C-rates (factor of charging power to nominal capacity). However, if we look at HPC (high power charging), we see that this assumption cannot be brought into agreement with reality [14].

HPC simulations often use data-driven models if measurement data are available [15]. Novel approaches follow AI or machines learning strategies [16,17]. When using data-driven models or AI algorithms, transfer to new battery or charging technologies requires more effort. Large amounts of data are again required for this. Therefore, an adaptation of these models is only possible under the condition of real data. [18]

Unsteady functions and models based on Kalman filters are also proposed [19,20]. The use of these models in current fleet management tools is only possible to a limited extent, as more complex functions would have to be incorporated here. This leads to an increased computing effort that cannot be handled by the current software solution. [21,22]

The methods mentioned here have useful applications for their respective fields of use. However, the methods described here are not sufficiently suitable for practical use in fleet management. A continuous simple function would be desirable for calculating the charging times and energy quantities as accurately as possible.

A major problem here is that the charging characteristics depend to a large extent on the battery management system (BMS) [23]. The BMS varies from vehicle to vehicle. In addition to the underlying battery technology (e.g., Li-NMC, LiFePo, …) differences in thermal management must also be taken into account [24]. It is, therefore, important for the calculations to be able to adapt the function to different vehicles. In this work, we search for a function to describe HPC operations that satisfy the following:

- Simple function with few parameters.
- The computational effort when using the method should be low.
- Easy adaptability to other BMS or alternative technologies.
- High accuracy.

The function is intended to provide further contributions in the following areas:

- Determination of exact charging times for given SOC rates and charging infrastructure solutions.
- Impact of superposition of multiple time-shifted charging events in terms of power consumption.
- Use in fleet management systems for the integration of e-trucks in logistics fleets.

### 1.2. Research Questions

This paper, therefore, deals with the problem of simulating the high-power charging of electric trucks. The following research questions were, therefore, formulated:

- How can HPC operations for e-trucks in terms of critical parameters (charging time, charging amount, SOC status) be calculated as easily as possible for fleet management tools?
- What accuracies can be achieved here compared to real-life data?
- Can the method be adapted to other charging characteristics?
- Which statements regarding the superposition of several charging processes can be derived from the model?

Based on real data of an electric truck field test, as well as on charging curves of established electric cars, an accurate method was developed to simulate charging processes. Since the size of the batteries and the charging performance varies depending on the application, the possibility was created to adapt this in the model.

*1.3. Logistic as a Function*

In one sentence, "The logistics mission is to provide the right quantity, of the right objects as objects of logistics (goods, people, energy, information), at the right place (source, sink) in the system, at the right time, in the right quality, at the right cost" [25]. Derived from this to the target of the logistic regarding truck fleets is as follows: It is to get goods to specific places at specific times and to do so at the lowest possible cost. The technology used, at least in relation to the truck, is reflected in the characteristics of the energy storage device for performing the work, in particular its type and capacity. The type of energy storage device is determined by the form of energy and its technology, e.g., diesel, $H_2$ or batteries. The required size of the energy storage is determined by the range and landscape to operate in and the weights of the transported goods.

Furthermore, the technology determines the type or location and the speed at which the truck's energy storage system can, should or must be refilled. The technology used also significantly determines the maintenance effort or the susceptibility to faults and the resulting costs. In order to illustrate a changeover between different technologies or to enable a comparison of these technologies, it is necessary to describe this formally. As described in [26], there are three main applications for a TCO (total cost of ownership) application. The three primary uses of TCO model data by the case study firms: supplier selection, supplier evaluation or measurement of ongoing supplier performance, and to drive major process changes/re-engineer. The TCO is, therefore, a good basis for an assessment. A detailed description of the TCO in transportation can be found in [27–29]. Therefore, the required TCO concept is presented below in a very condensed form.

Let TCO (I, E) be the mapping to determine the total cost of ownership, where I describe the total infrastructure (fuelling capabilities, trucks, and all downstream infrastructure required for operation and their expenses) and E describes the total energy consumption.

The required energy $E = \int P_t dt$ is determined by the load profile when refueling the truck fleet $P_t$. To design and determine $I\left(L_N, \vec{P_N}, P_{max}\right)$, we need $L_N$ (number and type of trucks), $\vec{P_N}$ (rated capacities of each fueling option), and $P_{max} = max(P_t)$ (maximum expected total fueling capacity). To determine $L_N\left(\vec{E_N}, \vec{P_N}; S_F, T_F, H_F\right)$, in addition to $\vec{E_N}$ (nominal capacities of the trucks' energy storage systems) and $\vec{P_N}$ (nominal capacities of the fueling infrastructure), we also need the mappings $S_F$ (service function of the fleet technology considered in the scenario), $T_F$ (trips to be made by the fleet), and $H_F$ (stopping times of the fleet for operation), which are very specific to the use case. Furthermore, for the dimensioning we still need, with $P_t\left(\vec{E_N}, \vec{P_N}; S_F, T_F, H_F\right)$, the load profile that the truck fleet causes during refueling. Since the load profile is generated by

$$P_t = \sum_{i=1}^{L_N} P_{t,i}, \tag{1}$$

it is often necessary to calculate the load profile for each truck when calculating the optimal fleet composition.

$S_F$, $T_F$, and $H_F$ remain predominantly unaffected by the choice of technology, as long as $H_F$ is only generated by normal operation and is not influenced by any refueling times $H_T$ via $\max(H_F, H_T)$. In any case, the main factors influencing the load profile during refuelling are $E_N$ and $P_N$, and the time required for refueling $H_T$ is also determined by these two variables. In conventional trucks, the refueling process does not happen during the truck loading times, but the time for refueling is short enough that it can be considered negligible. In the case of e-trucks, the refuelling times are naturally longer, but the refuelling process can be carried out during loading. If it is now possible to fill up the e-truck with sufficient energy in this time window, the refuelling times do not affect the stopping times. In the following section, an easy-to-use method is derived that can be used to solve the mentioned problems in a simple manner and with sufficient computing speed.

## 2. Materials and Methods

### 2.1. Data Basis

The measured data from the Megawatt Logistics project were used to create the model for the charging points. These are measured values at a 44 kW charging station from SCHRACK and a 150 kW charging station from ABB. The measurement is performed with a measuring rate of $dt = 2$ min. Via the charging stations, 26t electric trucks were loaded. The measurements were carried out over the course of the integration of the e-trucks into the running operation of logistics distribution centers.

Two different battery variants were used in the electric trucks. One version with 150 kWh installed battery capacity and 124 kWh usable capacity and one version with 224 kWh installed capacity and 184 kWh usable capacity. Due to the recording of the measured values during normal operation of the electric truck and the reduced usable battery capacity by the manufacturer, it was not possible to carry out series of measurements from SOC = 0 to SOC = 100. For this reason, those charging cycles were determined from the measured values which cover the largest possible SOC range. Typical power profiles are shown in Figure 1a for a 44 kW charging station and in Figure 1b for a 150 kW charging station. In the case of the 44 kW charging point, 39 out of 512 charging processes were found with a sufficiently large SOC range. In the case of the 150 kW charging point, 15 out of 104 charging processes were found.

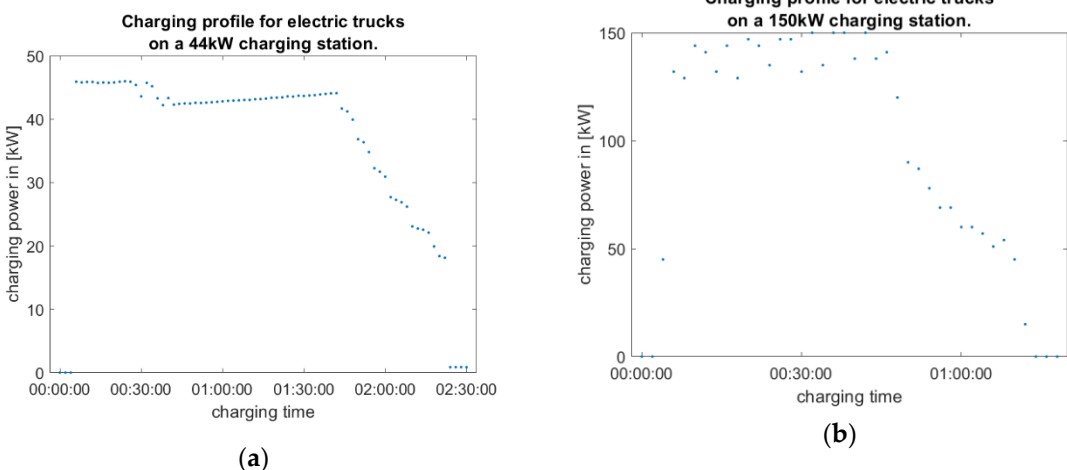

**Figure 1.** (**a**) Shows the individual measured values of an e-fueling of an e-truck with a battery capacity of 224 kWh at a charging station with 44 kW. (**b**) Shows the individual measured values of an e-fueling of an e-truck with a battery capacity of 224 kWh at a charging station with 150 kW.

### 2.2. Analytical Approach

The method described below for determining the charge power or state of charge of the battery has been developed for the following applications.

- Flexibility in terms of battery capacity.
- Flexibility with regard to the nominal capacity of charging stations.
- Determination of the instantaneous charging capacity of an electric truck.
- Determination of the state of charge of an electric truck.
- Applicability for the calculation of the characteristic values for the HUB design with regard to electric truck fleets.
- Applicability for the calculation of charging times for electric truck fleets.
- Determination of parameters for intelligent load management for electric truck fleets.

The following structure was defined in order to fulfil the goals of the model:

In the first step, a function $P_{t_0}(t; E_N, P_N)$ was used to describe the charging power away from the starting point $t_0$ at $SOC = 0$ with parameters $E_N$, the nominal capacity of the battery, and $P_N$, the nominal power of the charging station. Based on this, the current state of charge of the battery is calculated after the time t starting from a SCC = 0 at the beginning of charging via

$$E_{t_0} = \int_0^t P_{t0}(s; E_n, P_N)ds \tag{2}$$

Using these considerations, for the corresponding $P_N$ and $E_N$, the charging time $t_{SOC_x}$ can be calculated, which is needed, based on the model, to charge the battery from $SOC = 0$ to $SOC = x$. Using this time, we can now define the function

$$P_{t_{SOC_x}}(t; E_N, P_N) = P_{t_0}(t + t_{SOC_x}; E_n, P_n) \tag{3}$$

for the charge power, starting from any state of charge, and the function

$$E_{t_{SOC_x}}(t; E_n, P_N) \quad = E_{t_0}(t + t_0; E_n, P_N) - E_{t_0}(t_{SOC_x}; E_N, P_N) \\ = \int_0^t P_{tsoc_x}(s; E_N, P_N)ds \tag{4}$$

for the state of charge of the battery.

In order to be able to use the defined steps, we need a suitable function for $P_{t_0}$. To find this function we do not want to follow the typical path of piecewise functions but instead define the function using a closed formulation. By taking a look at the measured values from Figure 1a,b we can see that we need a function which is quite constant over a certain range and runs towards zero in a curve at the end. This behaviour can be achieved by a sigmoid function.

### 2.3. Identify $P_{t_0}$

As a test function, we choose $f(t) := e^{-(\lambda\, t)^k}$. This function is a function between 1 and 0, at $\lambda$ the function has a fixed value of about 0.63 and by k the speed is determined with which the curve goes towards zero. We use that the test function starts with one and scale the function with $P_N$. As a next step we perform an a priori conditioning of the shape of the curve using $\lambda := \frac{P_N}{E_N}$ and $k = 3 \cdot \frac{E_N}{P_N}$. From this we now obtain an estimate of the function

$$P_{t_0} = P_N \cdot e^{-\left(\frac{P_N}{f_{error}(P_N, E_N) \cdot E_N} \cdot t\right)^{3 \cdot \frac{E_N}{P_N}}} \tag{5}$$

with $f_{error}$ as a correction function to achieve the desired battery capacity. Actually, we would also need a correction function for the term at k. However, to calculate this, we need measured values from more than two different sizes of charging stations. With the value given for k we have a good approximation for the behavior at $P_N = 44$ kW. For more details see the point discussion.

### 2.4. Finding the Correction Function

To calculate the correction function, we continue as follows. In the first step we calculate the maximum reached energy capacity $E_{t_{end}}$ of the selected $E_N$ and $P_N$ using function (5). Based on the calculated values, a set of parameterized correction functions $f_{E_N}(P_N)$ each of the same type is determined. These functions calculate for each $E_N$ the error $\frac{E_N}{E_{t_{end}}}$ regarding $P_N$. In the next step we search for a homotopy [30] with $H(P_N, 0) := f_{E_{Nmin}}(P_N)$ and $H(P_N, 1) := f_{E_{Nmax}}(P_N)$. Our found homotopy H must also meet our functions $f_{E_N}$ for the chosen $E_{Nmin} < E_N < E_{Nmax}$. Through the transformation $(E_{Nmin} - E_N)/(E_{Nmax} - E_{Nmin})$ now follows the continuous deformation $H(P_N, E_N)$. This deformation now maps the individual functions $f_{E_N}$ into each other. The deformation H found in this way is our searched function $f_{error}$. Based on this we now get:

$$P_{t_0} = P_N \cdot e^{-\left(\frac{P_N}{H(P_N, E_N)\cdot E_N}\cdot t\right)^{3\cdot\frac{E_N}{P_N}}} \tag{6}$$

### 3. Results

### 3.1. Applying the Method

With regard to the battery capacities for electric trucks to be expected in the distribution logistics, the following nominal capacities $E_N \in \{600, 440, 360, 224, 150\}$ were selected for calculation. Additionally, the following nominal charging power $P_N \in \{600, 500, 360, 250, 150, 63, 44\}$ was selected. We now use the function defined in Equation (5) where, in the first step, we have chosen the correction function with $f_{error} = 1$. Below the results for $E_N = 224$ kWh and $E_N = 600$ kWh are shown in Figure 2a,b.

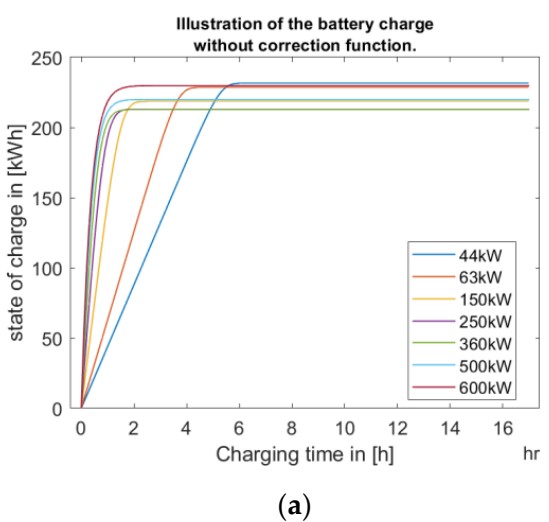
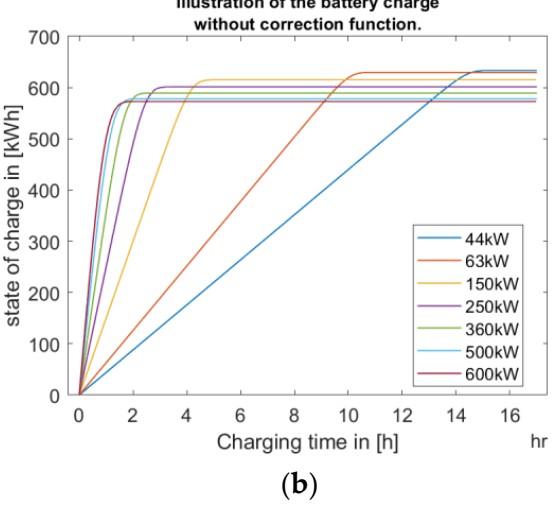

**Figure 2.** (**a**) Battery charge for 224 kWh battery with different charging powers without correction function; (**b**) battery charge for 600 kWh battery with different charging powers without correction function.

### 3.2. Calculate the Correction Function

In the first step, we calculate the errors $c_{P_N} = \frac{E_N}{\max(E_{t_0})}$, regarding $P_N$. From these values, we can calculate an error function with respect to $P_N$ for each $E_N$. In Figure 3a,b this error function is shown for $E_N = 224$ kWh and $E_N = 600$ kWh.

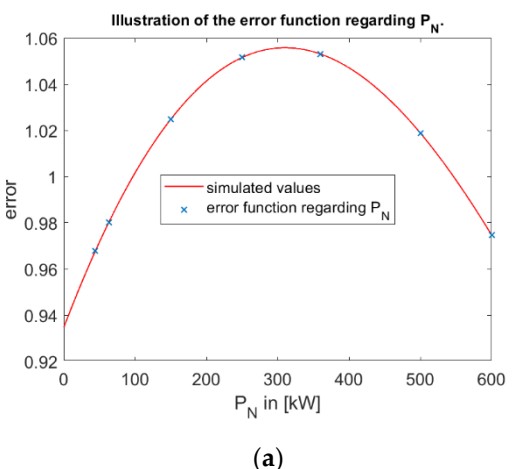

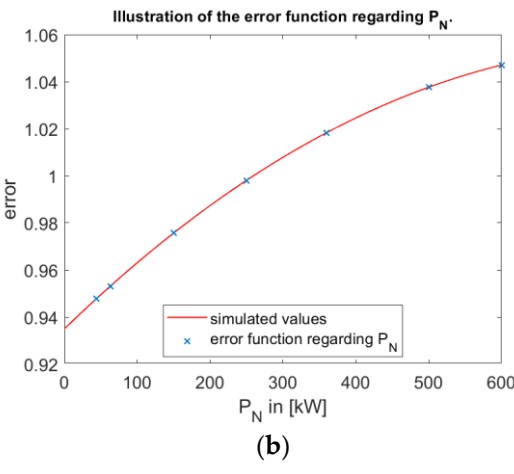

(**a**)

(**b**)

**Figure 3.** (**a**) Fitting of the error function, as a polynomial of degree 4, to the maximum state of charge of 150 kWh battery as a function of charge power. (**b**) Fitting of the error function, as a polynomial of degree 4, to the maximum state of charge of 600 kWh battery as a function of charge power.

The error $f_{error}(P_N; E_N)$ behaves for all $E_N$ fix as a fourth-degree polynomial with respect to $P_N$. So, we can define $f_{error}(P_N) = \sum_{i=0}^{M} a_i \cdot P_N^i$ with $M = 4$ for each $E_N$. Based on these functions we can now calculate a homotopy $H(P_N, E_N)$, which transfers each function into the other. We calculate this homotopy by a coefficient mapping. In Figure 4a,b the coefficient mapping is shown with respect to the coefficients $i = 0$ and $i = 4$. In Figure 4a,b the coefficient mapping was calculated using a polynomial of degree 4. The advantage for these procedures lies in the calculation of the error function via pure vector matrix operations. This results in a somewhat faster calculation than an exact approach. The disadvantage is that a sufficient accuracy is only given in a certain range. Fortunately, the accuracy can be calculated. This will be discussed in a later chapter.

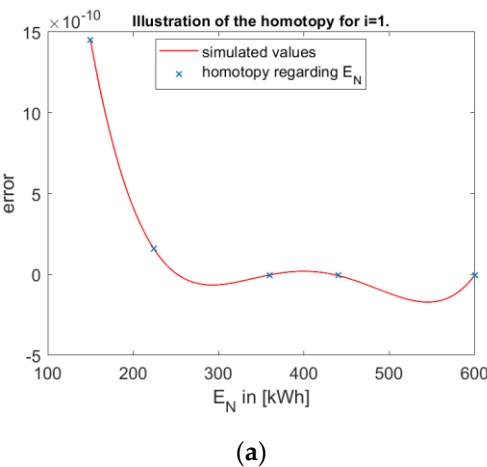

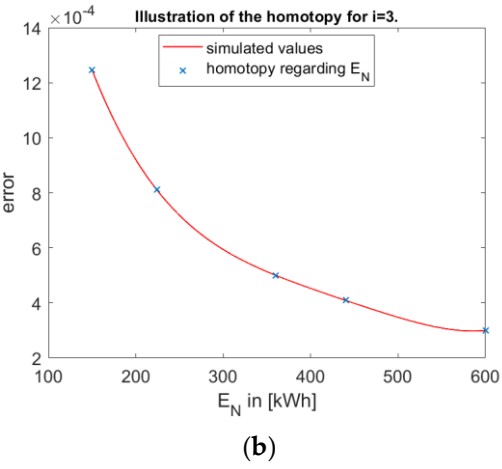

(**a**)

(**b**)

**Figure 4.** (**a**) Fitting the coefficient $i = 1$ of the charging power-correlated error function, as a polynomial of degree 4, as a function of battery capacity. (**b**) Fitting the coefficient $i = 3$ of the charging power-correlated error function, as a polynomial of degree 4, as a function of battery capacity.

In this step we now want to collect the first results of our work. We now insert our coefficient mapping and get

$$f_{error} = \sum_{i=0}^{M} \sum_{j=0}^{S} c_{i,j} \cdot E_N^j \cdot P_N^i \qquad (7)$$

Now, we get the representation for $H(P_N, E_N) = \vec{E}_N \times C \times \vec{P}_N$ as a vector matrix operation with $\vec{E}_N = \left(E_N^0, E_N^1, E_N^3, E_N^4\right)$, $\vec{P}_N = \left(P_N^0; P_N^1; P_N^2; P_N^3; P_N^4\right)$ and

$$
C = \begin{pmatrix}
-7.12 \times 10^{-23} & 5.57 \times 10^{-19} & -5.62 \times 10^{-16} & 7.57 \times 10^{-14} & -4.65 \times 10^{-13} \\
1.05 \times 10^{-19} & -9.19 \times 10^{-16} & 9.57 \times 10^{-13} & -1.32 \times 10^{-10} & 7.71 \times 10^{-10} \\
-5.23 \times 10^{-17} & 5.51 \times 10^{-13} & -6.01 \times 10^{-10} & 8.75 \times 10^{-8} & -4.65 \times 10^{-7} \\
9.03 \times 10^{-15} & -1.42 \times 10^{-10} & 1.68 \times 10^{-7} & -2.69 \times 10^{-5} & 1.21 \times 10^{-4} \\
-1.26 \times 10^{-13} & 1.33 \times 10^{-8} & -1.83 \times 10^{-5} & 3.81 \times 10^{-3} & 9.88 \times 10^{-1}
\end{pmatrix}
\tag{8}
$$

the coefficient matrix. We now get the following representation for

$$
P_{t_0} = P_N \cdot e^{-\left(\frac{P_N}{\vec{E}_N \times C \times \vec{P}_N \cdot E_N} \cdot t\right)^{3 \cdot \frac{E_N}{P_N}}}
\tag{9}
$$

### 3.3. Accuracy

In the following section, we want to discuss the achieved accuracy of our model. This is analysed from several perspectives. In a first step we want to see how well the SOC is reproduced with our model. For this purpose, we have calculated the SOC for different battery sizes with different nominal charging powers. To achieve this, we have used Equation (2) with Equation (9). In Figure 5a,b, we can see that we exactly reach the nominal capacity of the battery depending on the nominal charging power.

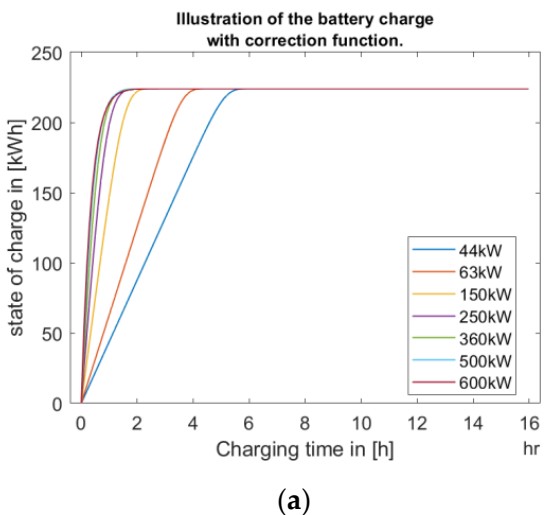

(a)

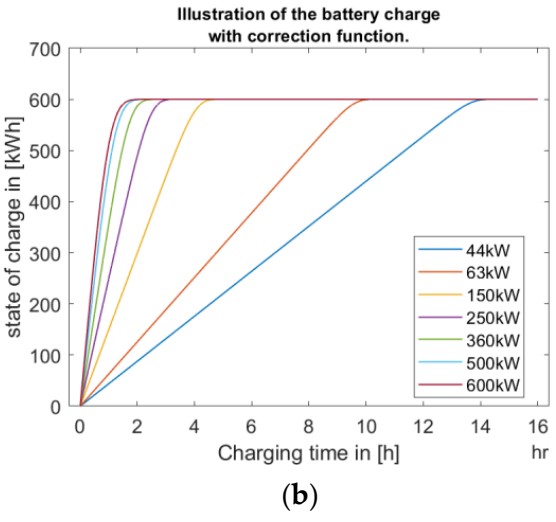

(b)

**Figure 5.** (**a**) Battery charge for 224 kWh battery with different charging powers and with correction function; (**b**) battery charge for 600 kWh battery with different charging powers and with correction function.

The question now is how well a nominal capacity is achieved, which lies between and outside the support points for calculating the correction function, Equation (7). The answer to this question is shown in Figure 6. Figure 6a shows the absolute error with the accuracy ranges of 1%, 2% and 5% with respect to the nominal battery capacity. In Figure 6b, the error is shown relatively. It should also be noted that the error regarding the battery capacity does not significantly change depending on the nominal charge capacity.

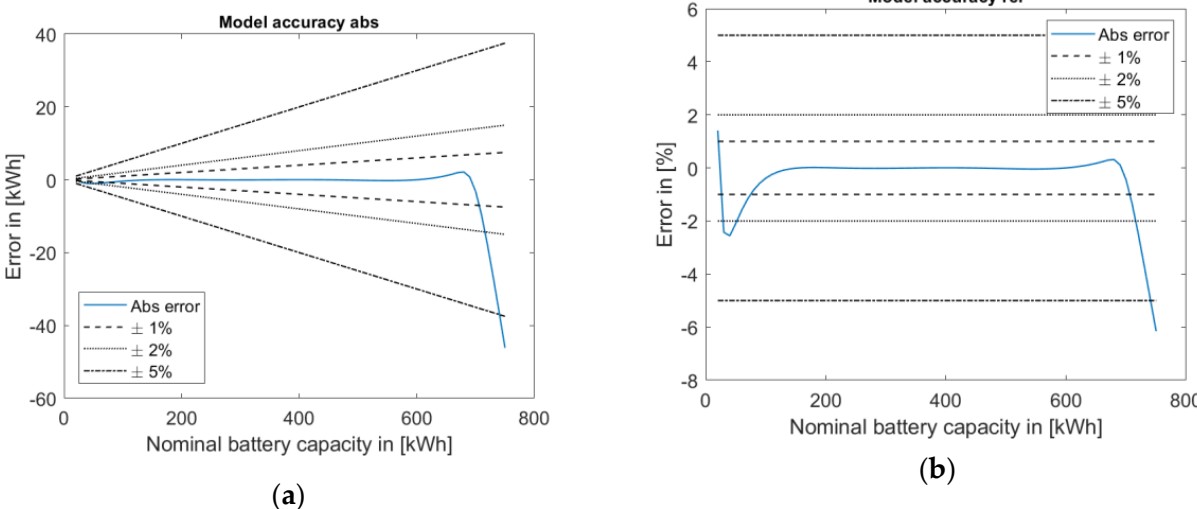

**Figure 6.** (**a**) Shows the absolute error with respect to the maximum capacity of the battery. (**b**) Shows the relative error with respect to the maximum state of charge of the battery.

The next question is how well the progression of the SOC is reproduced with this model. To answer this question, the results of the model were compared with measurement results from the ongoing operation of electric trucks. Due to the load management system and the limitations of the electric truck operation, it was not possible to measure the load curves in the area of 0% SOC. Figure 7a,b shows the result of the model regarding the state of charge of the battery compared to the measured values at the charging station.

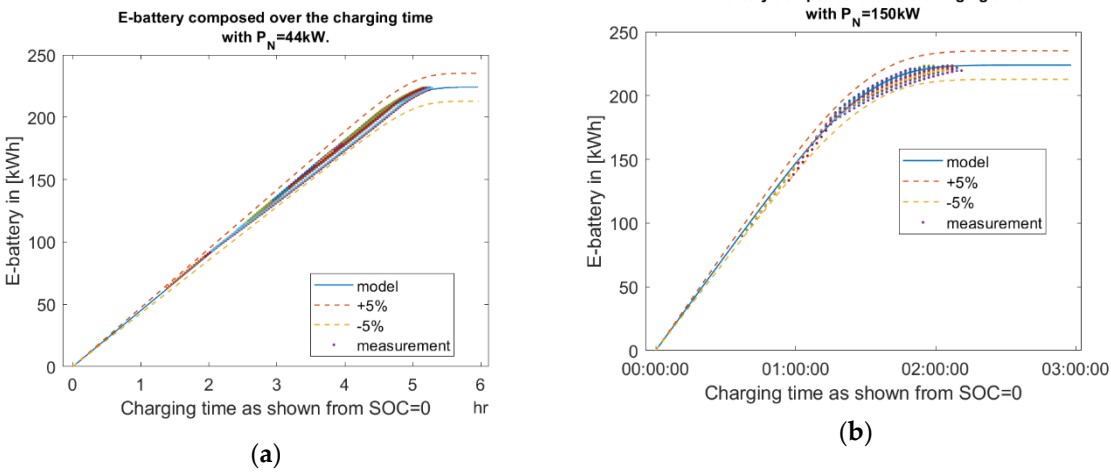

**Figure 7.** (**a**) Shows the calculated progression of the state of charge of an e-truck with 224 kWh battery capacity at 44 kW charging point with the series of measurements overlaid. (**b**) Shows the calculated progression of the state of charge of an e-truck with 224 kWh battery capacity at 150 kW charging point with the series of measurements overlaid.

The last question we consider is the accuracy of the current power $P_{t_{SOC_x}}$. This means how well the model describes the instantaneous power of the charging column after time t with an initial SOC $=$ x. Without loss of generality, we can limit this question to the case SOC $=$ 0. Figure 8a,b shows the measured values and the model. As we can see, the approximation for the version with $P_N = 44$ kW is good. If a higher accuracy is required for the range of higher $P_N$, the model can be improved by further measurements in the range of higher $P_N$, as described in the methodology part.

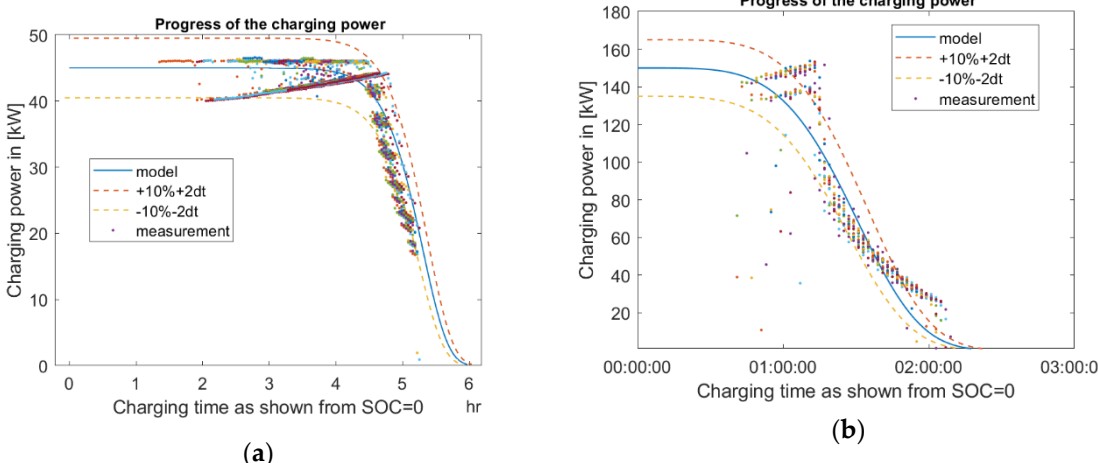

**Figure 8.** (**a**) Shows the calculated characteristic of the charging power of an e-truck with 224 kWh battery capacity at 44 kW charging station with the series of measurements overlaid. (**b**) Shows the calculated characteristic of the charging power of an e-truck with 224 kWh battery capacity at 150 kW charging station with the series of measurements overlaid.

### 3.4. Simulation Results

In the following, an application of the above developed method to describe e-charging processes of several e-trucks is displayed. As a result, the power simultaneity of super-positioned charging operations are presented. The actual power load on the transformer in a scenario with N = 10 e-trucks was calculated. In Figure 9, cases for a usable battery capacity of $E_N$ = 182 kWh and the nominal power of the charging stations $P_N$ = 44 kW and $P_N$ = 350 kW and with a SOC = 0% or SOC = 50%. Figure 9a shows the transformer power for an initial $SOC_{t_0}$ = 0%, for equidistant arrival times of dt = 5 min, dt = 15 min and dt = 30 min. Figure 9b shows the transformer power of the same setting but with an initial $SOC_{t_0}$ = 50%.

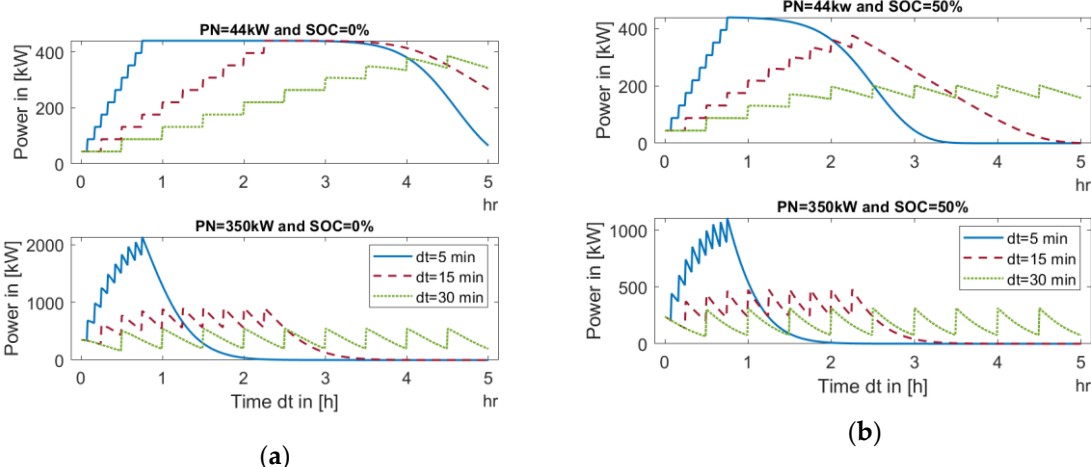

**Figure 9.** (**a**,**b**) Shows the interaction of SOC level, charging station power and the time interval of the arrivals with respect to the simultaneity ratio for 10 e-trucks with 224 kWh battery capacity.

In these four scenarios, it is easy to see that the maximum power demand does not necessarily form at the full possible amount. It can also be clearly seen that the power maximum depends on the parameters dt and the initial $SOC_{t_0}$. It is also obvious that this dependence changes with respect to the nominal power of the charging point $P_N$. Now, the

relevant question is the change in this maximum with respect to $SOC_{t_0}$ and $P_N$ in relation to dt. To address this question, we use the definition of the diversification factor

$$f_d = \frac{\max(\sum_{i=1}^{N} P_i(t))}{\sum_{i=1}^{N} P_{N_i}} \tag{10}$$

with $P_i(t)$ as the current power at the i-th charging point and $P_{N_i}$ as the nominal power at the i-th charging point. In Figure 10a, $f_d(dt, P_N; SOC_{t_0}, E_N)$ is shown, and in Figure 10b, $f_d(dt, SOC_{t_0}; P_N, E_N)$ is shown. As can be seen in the figures, $f_d$ decreases faster as a function of $SOC_{t_0}$ than as a function of $P_N$ with respect to dt.

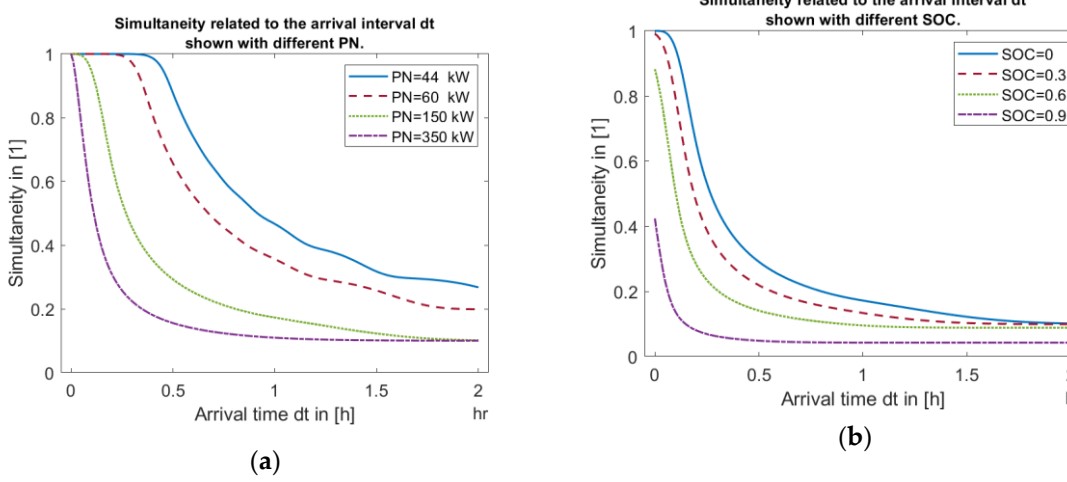

**Figure 10.** (**a**) The influence of the time interval on the simultaneity for 10 e-trucks with a battery capacity of 224 kWh and a SOC = 0 with respect to different charging powers. (**b**) The influence of time interval on simultaneity for 10 e-trucks with a battery capacity of 224 kWh and a charging power of 150 kW with respect to different SOC.

## 4. Discussion

With this work, a new method was presented to simulate charging processes in the field of HPC for e-vehicles. As shown, this can significantly contribute to the conversion towards e-mobility in the field of logistics. With the help of the described function, basic considerations as well as specific design calculations for the required charging infrastructure are possible. Applications in the logistics sector can be checked for a possible conversion in the direction of e-mobility. With reference to the research questions formulated at the beginning, the following aims were achieved:

### 4.1. Use Case E-Trucks

Based on the formulations in Section 1.3 a method for determining the necessary infrastructure for logistics hubs was created using the mapping presented in Equation (9) for the charging process via a simulation. Here, in the first step, $L_N$, $P_t$ and $F_N$ are determined via probabilistic optimization. Where $F_N$ denotes the number of charging points, by using Equation (1), $F_N$ can be calculated, as given in Equation (11)

$$P_t = \sum_{i=1}^{L_N} P_{t,i} = \sum_{j=1}^{F_N} P_{t,j} \tag{11}$$

For this purpose, the distribution densities for the stopping times were determined from the data of the corresponding hubs, and a set of the necessary random variables for $H_F$ was generated. From the available data, the tours and their composition were extracted. In the process, corresponding delivery windows were also described. Due to the lack of available data, $S_F$ was not yet simulated in this simulation setup, but a necessary reserve infrastructure was estimated. Now, using the random variables for $H_F$ and a random

assignment of the tours $T_F$, the operation of the hub was often sufficiently simulated while maintaining the delivery windows. Here, the choice of possible values for $\overrightarrow{P_N}$ and $\overrightarrow{E_N}$ remained fixed, but not their number. This probabilistic optimization was now repeated for different but selected sets of values of $\overrightarrow{P_N}$ and $\overrightarrow{E_N}$. On this basis, the TOC of the fleet could thus be estimated.

It has been shown that the calculation of the refuelling process plays a time essential role and, with the above presented Formula (9), a sufficient speed of the calculation could be achieved. It also showed that a closed formulation regarding the two variables $P_N$ and $E_N$ was sufficient and very helpful.

### 4.2. Gradient

As mentioned in Section 3.2, the gradient of the load function is only adjusted in two points via the exponent k in Equation (5). We have used a linear function for this. It has been shown that, when using a constant to describe the slope, means for each combination of $P_N$ and $E_N$ in Equation (5) the same value then the correction function (7) is equal to one and is, therefore, not needed. For a nonlinear function, finding the correction function becomes a bit more complex.

### 4.3. Accuracy

Since, in our application, the computational speed was more critical than the validity range of the calculation (see Figure 4), in the second step of the calculation of the correction function (7) a polynomial of degree 4 was also used. In this step, an approach with exponential functions $c_i = a * e^{b*x} + c * e^{d*x}$ would produce a larger validity range, see Figure 11.

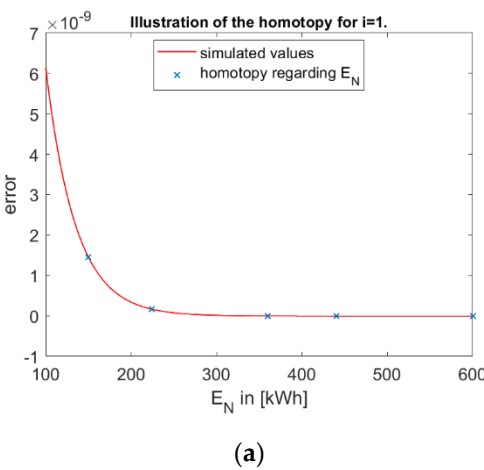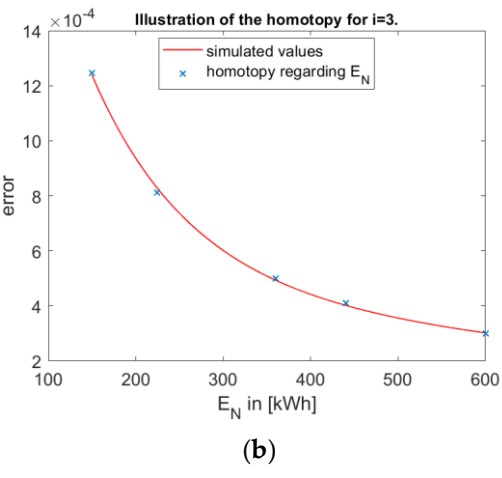

(a)

(b)

**Figure 11.** (**a**,**b**) A better fit of the error function by using an exponential model. This also shows an applicability with this approach to a larger domain (compare to Figure 4).

However, a vector-valued function would be needed in the calculation of the correction function and could not be represented purely by vector and matrix multiplications. This is unproblematic with few computations, but with our approach and the associated large number of repeated computations over time was, in fact, problematic. However, the validity range can be extended over the approach shown above.

### 4.4. Superposition of Several Charging Processes

With the method described in this paper, it could be shown that it is possible to represent charging processes of batteries in the e-mobility sector in HPC by means of a parametrized continuous function. This is necessary for many applications in the area of determining the simultaneity of several e-vehicles for the design of the power electronics.

In comparison with real data, it could be shown that the power consumption over time, as well as the calculated SOC of the batteries, can be very well described with the model.

On the basis of superpositions of 10 simulated charging processes, it was possible to calculate the simultaneity and thus the required power electronics. This showed that minimization was very simple, especially for very high charging powers (150 kW, 350 kW). A time shift of only 15 min between the start of the individual charging processes is sufficient to achieve a reduction in the simultaneity to only 0.3 (case 350 kW). The reason for this is the rapid reduction in the instantaneous charging power during HPC.

The second important quantity for the calculation of the concurrency is the starting SOC of the battery. If the battery is still relatively full (SOC > 0.6), this also shows a low simultaneity.

These calculations are indispensable, especially for calculations of the required infrastructure of charging solutions for many vehicles with HPC.

It was also shown that, at low charging powers ($P_N < 44$ kW), the simultaneity is higher from the ground up. This is due to charging at a lower C-rate, as it is possible to charge at the rated power for a longer period of time.

## 5. Conclusions

The goal of this work is to provide a method to describe HPC operations for fleet management systems. Thereby, a method was developed to approximate the charging process with a continuous function.

The function, once parameterized, can be used to quickly calculate the required standing times of e-trucks. Direct integration into existing dispatching tools is possible and envisaged.

It is possible to calculate charging times and energy quantities with sufficient accuracy. Particularly for the management of entire e-fleets, this is important and can be an essential building block for the future integration of e-vehicles in logistic operations.

For future new battery and BMS technologies, adaptations of the model can be easily made. Adaptations are easy to implement here and should also help to provide good forecasts for charging characteristics in the future. The function can also be used for other alternative technologies, such as FCEV (fuel cell electric vehicle) apart from BEV (battery electric vehicle). For this purpose, only the basic parameters have to be adapted.

The associated increase in sustainability through alternative fuels in the logistics sector is to be supported and pushed with this method.

In the field of urban logistics, a simple method for calculating the required standing times can now be used for dispatching. Due to the simple structure, use within the current dispatching tools is possible.

**Author Contributions:** Conceptualization, T.M. and D.W.; methodology, T.M.; software, T.M., P.S.; validation, T.M., D.W.; formal analysis, T.M.; investigation, D.W.; resources, W.M.; data curation, T.M.; writing—original draft preparation, T.M. and D.W.; writing—review and editing, W.M., D.W., T.P.; visualization, T.M.; supervision, T.P.; project administration, W.M.; funding acquisition, W.M. All authors have read and agreed to the published version of the manuscript.

**Funding:** This work was funded in the Programme Line "Austrian Electric Mobility Flagship Projects-9th call"—An initiative of the Federal Ministry for Transport, innovation and Technology (Nr.: 867706). We gratefully acknowledge financial support from the Austrian Ministry for Transport, Innovation and Technology and the Austrian Research Promotion Agency (FFG), the Austrian Climate and Energiefund (KLIEN) and the Kommunalkredit Public Consulting GmbH (KPC).

**Institutional Review Board Statement:** Not applicable.

**Informed Consent Statement:** Not applicable.

**Data Availability Statement:** Additional data can be found at [9].

**Acknowledgments:** Open access funding provided by BOKU Vienna Open Access Publishing Fund.

**Conflicts of Interest:** The authors declare no conflict of interest. The funders had no role in the design of the study; in the collection, analyses, or interpretation of data; in the writing of the manuscript, or in the decision to publish the results.

## Nomenclature

| | | |
|---|---|---|
| HPC | high-power charging | |
| SOC | state of charge | |
| LCA | life cycle assessment | |
| C-rates | factor of charging power to nominal capacity | $[h^{-1}]$ |
| AI | artificial intelligence | |
| TCO | total cost of ownership | |
| $TCO(.,.)$ | function for the TCO | |
| E | energy | |
| $I, I(.,.,.)$ | mapping of the total infrastructure | |
| $P_t, P_t(.,.;.,.,.)$ | power at time t | [kW] |
| $L_N, L_N(.,.;.,.,.)$ | mapping for the number and types of trucks in the fleet | |
| $\vec{P_N}$ | rated capacities of each fuelling option | [kW] |
| $P_{max}$ | maximum expected total fuelling capacity | [kW] |
| $\vec{E_N}$ | nominal capacities of the trucks' energy storage systems | [kWh] |
| $T_F$ | trips to be made by the fleet | |
| $H_F$ | stopping times of the fleet for operation | [s] |
| $H_T$ | refuelling times | [s] |
| $S_F$ | service function of the fleet technology considered in the scenario | |
| dt | Timestep | [s] |
| $P_{t_0}(.;.,.)$ | Function to describe the charging power away from the starting point $t_0$ at SOC $= 0$ | [kW] |
| $t_0$ | time at SOC $= 0$ | [s] |
| $E_{t_0}$ | function to describe battery capacity at time t from start at $t_0$ | [kWh] |
| $P_{t_{SOC_x}}(.;.,.)$ | charging power at time t from start at SOC $= x$ | [kW] |
| $E_{t_{SOC_x}}(.;.,.)$ | function to describe battery capacity at time t from start at SOC $= x$ | [kWh] |
| $f(.)$ | test function | |
| $f_{error}(.,.)$ | correction function to achieve the desired battery capacity | |
| $E_{t_{end}}$ | the calculated maximum reached energy capacity | [kWh] |
| $H(.,.)$ | homotopy | |
| $f_{E_N}(.)$ | parameterized correction function | |
| $f_{E_{N_{min}}}(.)$ | lowest parameterized correction functions | |
| $f_{E_{N_{max}}}(.)$ | highest parameterized correction functions | |
| $E_{N_{min}}$ | lowest nominal capacities used | [kWh] |
| $E_{N_{max}}$ | Highest nominal capacities used | [kWh] |
| $c_{\vec{P_N}}$ | correction coefficients | |
| $\vec{E_N}$ | vector with powers of $E_N$ | |
| $\vec{P_N}$ | vector with powers of $P_N$ | |
| $f_d, f_d(.,.;.,.)$ | diversification factor | |
| $P_i(.)$ | current power at the i-th charging point | [kW] |
| $P_{N_i}$ | is the nominal power at the i-th charging point | [kW] |

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
