# Peer review of "Novel Modelling Approach for the Calculation of the Loading Performance of Charging Stations for E-Trucks to Represent Fleet Consumption"

_energies, doi:10.3390/en14123471_

Round 1

Reviewer 1 Report

The subject of this paper is up-to-date. However, after a careful reading of this article, the following issues were formulated. 

- The reader does not know at first what the purpose of the article is; It is also not clearly shown when reading it; 

- line 30: "LCA" - there is no explanation of this abbreviation; 

- line 53: "C-rates" - there is no explanation; 

- section "1.1 Literature Review" - the literature review is very poor and unclear; it is recommended to deepen and present the current state of knowledge in the problem much better;

- section 1.3 comes before section 1.2 - is it a mistake in the numbering or is the text order changed?

- Figures: the points on the charts are difficult to see because the charts are small and the points are small and light in color; it is recommended to improve the graphs of these charts to make them easier to read; 

- all formulas use the assignment operator ": =", which is used in codes or pseudocodes. However, the correct mathematical operator should be the sign "="; It is therefore advisable to improve the formulas in this respect; 

GENERAL Note:

The article requires major rewriting and supplementation, as the current order of the content makes the article unclear. First, the research problem should be presented along with the justification for its research. Next, a thorough literature review should be made on the current state of knowledge regarding the research problem. Then, the methodology of the research should be presented, followed by the basic research results along with their critical analysis and evaluation - taking into account the comparison of the obtained results with similar results, included in other similar studies.  

Author Response

Thank you very much for the review. The comments were important to move the paper forward. We think we have provided more clarity with the changes regarding the contribution of the paper. You can find the answers to your review in the attached file.

Reviewer 2 Report

The subject of the manuscript is interesting. However, there are few concerns and comments as listed below:

- The English and grammar of the manuscript need modification. There are typos and grammatical errors. The structure and format of the paper need to be modified. It is difficult to follow the paper in the present format and how it is explained and presented. The sentences may need to be written in a formal (professional) format. For example, the first paragraph in section 3.4 is unclear.

- In section 1.3 and 4.1 the equations need to be written in the right format. Please provide numbers for equations and refer to the correct formatting of the journal. If the equations are coming from a reference, please provide the reference.

- Again, in this section, line 293 authors explain “ In these four scenarios it is nice to see that a maximum is formed”. They  need to be be explained completely to ensure the readers completely understand the graphs and simulation results. “Maximum of what??”

- Conclusion is incomplete and does not show the contribution of the paper. Abstract also needs to represent the contribution of the paper.

- Authors are requested to exactly mention what they are looking for by this modelling approach and maximizing power of charging station, size of the battery and SOC. Authors are suggested to add the main contributions in dot points in introduction section.

Author Response

(The authors gave the same response as above.)

Round 2

Reviewer 1 Report

The changes made to the text are satisfactory. 

Reviewer 2 Report

Authors have successfully addressed the comments.